# An mRNA Vaccine for Herpes Zoster and Its Efficacy Evaluation in Naïve/Primed Murine Models

**DOI:** 10.3390/vaccines13030327

**Published:** 2025-03-19

**Authors:** Linglei Jiang, Wenshuo Zhou, Fei Liu, Wenhui Li, Yan Xu, Zhenwei Liang, Man Cao, Li Hou, Pengxuan Liu, Feifei Wu, Aijun Shen, Zhiyuan Zhang, Xiaodi Zhang, Haibo Zhao, Xinping Pan, Tengjie Wu, William Jia, Yuntao Zhang

**Affiliations:** 1CNBG-Virogin Biotech (Shanghai) Co., Ltd., Shanghai 201800, China; jianglinglei@126.com (L.J.); zhouwenshuobiology@gmail.com (W.Z.); liufei@cnbg-virogin.com (F.L.); liwenhui@cnbg-virogin.com (W.L.); xuyan@cnbg-virogin.com (Y.X.); liangzhenwei@cnbg-virogin.com (Z.L.); caoman@cnbg-virogin.com (M.C.); houli@cnbg-virogin.com (L.H.); liu_pengxuan@163.com (P.L.); wufeifei@cnbg-virogin.com (F.W.); shenaijun@cnbg-virogin.com (A.S.); 13086698536@163.com (Z.Z.); zhangxiaodi@cnbg-virogin.com (X.Z.); zhaohaibo@cnbg-virogin.com (H.Z.); panxp@cnbg-virogin.com (X.P.); wutengjie@cnbg-virogin.com (T.W.); 2Shanghai-Virogin Biotech Co., Ltd., Shanghai 201800, China; 3Sinopharm Group China National Biotech Group (CNBG) Co., Ltd., Beijing 100124, China; 4State key Laboratory of Novel Vaccines for Emerging Infectious Diseases, China National Biotech Group (CNBG) Co., Ltd., Beijing 100024, China

**Keywords:** herpes zoster, mRNA vaccine, immunogenicity, intradermal administration, lyophilization

## Abstract

**Background/Objectives:** An overwhelming burden to clinics, herpes zoster (HZ), or shingles, is a painful disease that occurs frequently among aged individuals with a varicella-zoster virus (VZV) infection history. The cause of shingles is the reactivation of dormant VZV in the dorsal root ganglia/cranial nerves of the human body. Patients with HZ experience sharp, intense, electric shock-like pain, which makes their health-related quality of life (HRQoL) extremely low. **Methods:** Various mRNA constructs were designed based on intracellular organelle-targeting strategies and AI algorithm-guided high-throughput automation platform screening and were then synthesized by in vitro transcription and encapsulated with four-component lipid nanoparticles (LNPs). Immunogenicity was evaluated on a naïve mouse model, long-term mouse model, and VZV-primed mouse model. Safety was evaluated by a modified “nestlet shredding” method for potential adverse effects induced by vaccines. Comparison between muscular and intradermal administrations was conducted using different inoculated approaches as well. **Results:** The best vaccine candidate, CVG206, showed robust humoral and cellular immune responses, durable immune protection, and the fewest adverse effects. The CVG206 administered intradermally revealed at least threefold higher humoral and cellular immune responses compared to intramuscular vaccination. The manufactured and lyophilized patch of CVG206 demonstrated good thermal stability at 2–8 °C during 9 months of storage. **Conclusions:** The lyophilized mRNA vaccine CVG206 possesses remarkable immunogenicity, long-term protection, safety, and thermal stability, and its effectiveness could even be further improved by intradermal administration, revealing that CVG206 is a promising vaccine candidate for HZ in future clinical studies.

## 1. Introduction

Herpes zoster (HZ), or shingles, is a painful disease caused by the reactivation of varicella-zoster virus (VZV), the same virus that also causes chickenpox. VZV enters the human body through the mucosal epithelial sites of the upper respiratory tract and spreads to the tonsils and other regional lymphoid tissues of Waldeyer’s ring, where VZV gains access to T cells [1,2]. VZV is then delivered to cutaneous sites and further transported to neuronal nucleic by viremia, then establishes latency in dorsal root ganglia after primary infection [3]. When the host’s immunity decreases, most frequently observed in elderly people or HIV-infected individuals, VZV can be reactivated and cause a blistering rash on the skin. The most common complication of HZ is persistent, unbearable postherpetic neuralgia (PHN) [4]. Globally, the cumulative incidence of herpes zoster is 2.9 to 19.5 per 1000 population, and is higher for females (3.22–11.2 per 1000 population) than males (2.44–8.0 per 1000 population) [5]. In Asian countries, especially Japan and South Korea, the incidence rate is overwhelmingly high and continues to increase [6]. In China, the total rate is 2.9 to 5.8 per 1000 annually, and is higher for females (3.94–7.9 per 1000 population) than males (2.86–7.6 per 1000 population) [7]. The cumulative incidence of HZ in North America is approximately 5.49–8.67 per 1000 annually and 5.77–9.85 per 1000 in Europe [5].

The advent of safe and effective anti-varicella-zoster virus (VZV) therapies has markedly reduced morbidity and mortality associated with varicella and herpes zoster (HZ), particularly among immunocompromised populations [8]. Current standard treatments, such as nucleoside analogues (e.g., acyclovir, valaciclovir), inhibit viral DNA polymerase, but face limitations including rising drug resistance (observed in up to 27% of immunocompromised patients), inadequate prevention of postherpetic neuralgia (PHN, which affects 20%–40% of cases), and dependence on renal function-adjusted dosing regimens [9,10,11]. While newer agents like amenamevir, a helicase–primase inhibitor with once-daily dosing, offer convenience, their restricted efficacy against PHN and localized safety concerns have hindered broader clinical adoption [11].

Vaccination strategies have advanced prevention, exemplified by two representative VZV vaccines that are currently available on the market: the live-attenuated Zostavax^®^ developed by Merck & Co. and the adjuvanted subunit vaccine Shingrix^®^ developed by GlaxoSmithKline. Zostavax^®^ was approved by the FDA in 2006, and its protective efficacy was 51% for shingles and 67% for PHN among participants over 60 years of age. Shingrix^®^ was approved by the FDA in 2017 [12], with an effectiveness of 97.2% in preventing shingles among participants over 50 years old [13]. The main components of Shingrix^®^ are a truncated VZV envelope glycoprotein (gE) and AS01B adjuvant system, which can enormously increase the immunogenicity of gE. The critical component of the AS01B system is Quillaja saponaria Molina fraction 21 (QS21), which derives from the bark of the Quillaja Saponaria Molina tree, a plant native to Chile [14,15]. However, challenges persist for both these commercial vaccines: Zostavax is contraindicated in immunocompromised groups, and the effectiveness of Zostavax^®^ waned substantially over time following vaccination [16] and eventually was withdrawn from the market on 18 November 2020 in the United States [17]. Shingrix^®^ requires two doses and triggers systemic adverse reactions (e.g., fever, myalgia) in over 30% of recipients [18,19]. Challenges also exist in production, quality control, stability, toxicity, and undesirable hemolytic effects when using the QS21-based system [20].

These shortcomings underscore the demand for novel solutions. mRNA vaccines, designed to encode multi-antigen targets (e.g., glycoprotein E, capsid proteins), hold promise by eliciting robust CD8+ T cell responses to eliminate latent virus, potentially lowering PHN risk, and present safer administration for immunocompromised individuals and rapid adaptability to viral evolution [21,22]. mRNA technology offers many unique advantages over traditional methods as a cutting-edge platform in the new era of vaccine development. Unlike inactivated, live-attenuated, and subunit-recombinant vaccines that rely on proteins or the entire pathogen as inoculants, mRNA vaccines directly modify and deliver mRNA molecules encoding appropriate antigen(s) as the active agent. This approach bypasses issues such as fusion protein folding, loss of exogenous genes, toxicity of viral vectors, and low immunogenicity in the absence of appropriate adjuvants. The versatility of mRNA technology is particularly beneficial for intracellular bacterial organisms with complex genomes. The ability to manipulate mRNA sequences provides flexibility, unparalleled design capabilities, and a high throughput for antigen and vaccine candidate trials. The mRNA molecule itself can act as a self-adjuvant, stimulating Toll-like receptors 3, 7, and 8 and inducing robust humoral and cellular immune responses [23]. This is because of the ability to specifically target CD4+/CD8+ T cells based on the customized mRNA sequence. mRNA technology entered the spotlight during the pandemic and has achieved tremendous success. Two well-known SARS-CoV-2 mRNA vaccines, Comirnaty^®^ (developed by Pfizer/BioNTech) and Spikevax^®^ (developed by Moderna), were approved by the FDA to restrain this pathogen spreading [24]. Notably, Moderna only took 42 days to design and produce a good manufacturing practice (GMP) standard SARS-CoV-2 mRNA vaccine for clinical trials [25], benefiting from the incredible advantages of mRNA vaccine technology over traditional vaccine platforms. Although the novel SARS-CoV2 mRNA vaccines are associated with certain adverse effects, advancements in RNA vaccine technology hold significant promise for mitigating and potentially resolving these challenges. Importantly, the observed risks are acceptable trade-offs when weighed against the substantial advantages of this technology. The primary causes of adverse reactions to COVID-19 mRNA vaccines are as follows. Firstly, liver-targeting effects of lipid nanoparticles (LNPs) exist. LNP accumulation in the liver is due to the organ’s unique sinusoidal vasculature, which promotes nanoparticle uptake and causes hepatic accumulation. This may induce transient local inflammation and elevated liver enzymes (e.g., ALT/AST). The ionizable lipids within LNPs (e.g., ALC-0315) can activate immune cells, triggering the release of proinflammatory cytokines (e.g., IL-6) that indirectly contribute to hepatocyte stress. Due to individuals’ metabolic variability, those with pre-existing liver dysfunction exhibit reduced capacity to clear LNPs or inflammatory mediators, increasing susceptibility to cumulative toxicity [26]. Secondly, PEG-mediated allergic reactions (rare) were observed. Polyethylene glycol (PEG), a component of LNPs, may rarely induce IgE-mediated hypersensitivity. Notably, PEG is widely used in consumer products (e.g., cosmetics and skincare), leading to pre-existing anti-PEG antibodies in some individuals. These antibodies heighten the risk of post-vaccination allergic responses [27]. Finally, localized immune reactions occur. Transient injection-site pain or swelling arises from LNP- or mRNA-driven activation of local immune cells (e.g., macrophages), which release inflammatory mediators such as histamine. To overcome these shortcomings, hepatic exposure can be mitigated through several approaches. LNP design parameters and component ratios can be optimized to enhance target specificity, tissue-specific delivery vectors targeting extrahepatic sites, such as skeletal muscle, could be engaged. PEG-free LNPs or alternative stabilizers to reduce allergenic potential can be developed, and individualized vaccination protocols that adjust dosage regimens for patients with pre-existing hepatic conditions or immunocompromise should be implemented. Complementary to these interventions, systematic hepatic function monitoring should be integrated throughout the vaccination process, incorporating baseline assessments and post-administration surveillance to ensure therapeutic safety. The mRNA vaccine platform elucidates convenient, flexible, and versatile vaccine antigen design, the robust cellular immune response induced by exogenous mRNA, self-adjuvant lipid nanoparticles (LNP) causing an innate immune response, and the ability to be easily scaled up for massive manufacturing. Most recently, Pfizer/BioNTech initiated a phase I/II clinical study of VZV mRNA vaccine candidates in February 2023 [28]. Moderna’s mRNA-1468 has initial data from phase I/II of the study in March 2024 [29]. Greenlight Biosciences, China Pharmaceutical University, and Innorna have finished a proof-of-concept study of their VZV mRNA vaccines [30,31,32].

We also selected the VZV gE protein as the antigen because of its high abundance and immunogenicity. In the present study, we tested the hypothesis that truncated gEs may result in different immunogenicity. Accordingly, different mRNA sequences encoding antigens were designed. Humoral and cellular immune responses were evaluated systematically in naïve and VZV-primed mouse models. We selected a full-length gE, CVG206, as the antigen encoded by an optimized mRNA sequence. The sequence of mRNA-1468 developed by Moderna encodes a 573AA-truncated gE protein with a Y569A mutation to facilitate the trafficking of proteins to the trans-Golgi network (TGN) [21]. We included this as a benchmark in our study (CVG200). Our study demonstrates that CVG206 has a good safety profile, triggering strong humoral and cellular responses, and importantly, this lyophilized mRNA vaccine shows potential to induce a long-term immune response, which is promising for future clinical investigation.

## 2. Materials and Methods

### 2.1. Generation of Plasmid Templates

The coding sequences (CDSs) described in 1.1 were based on the sequence of the VZV vOKa strain (GenBank: AB097932.1), and codon optimization was performed using the Gensmart™ Codon Optimization Tool. The CDS fragments were synthesized by GenScript Biotech Corporation and subcloned into CNBG-Virogin’s proprietary plasmid template by homologous recombination. All constructs were verified with the expected length of the poly-A tail.

### 2.2. Plasmid Linearization and In Vitro Transcription (IVT)

Each plasmid template was linearized using 2 μg plasmid with 1 unit of BspQI (Hongene Biotech (Shanghai, China), ON-124-01) in 1x cut buffer (Hongene Biotech, ON-268). The linearization reaction was incubated in a 50 °C water bath for 1.5 h. The linearized products were subjected to 0.8% agarose gel electrophoresis analysis. The purification of linearized products was performed with a DNA purification kit (GeneOn BioTech (Brno, Czech Republic), PCRF) in an automatic nucleic acid isolation instrument (Hanweisci (Ningbo, China), HW-24L).

“One-pot” in vitro transcription was used for mRNA production. All reagents were purchased from Hongene Biotech. In the reaction mix, the concentration of ATP, CTP, GTP, N1-Me-pUTP, and (3′-OMe-m7G) (5′) PPP (5′) (2′-OMeA) pG was 8 mM, the linearized plasmid concentration 0.05 μg/μL, inorganic pyrophosphatase (iPPase) 0.001 unit/μL, T7 RNA polymerase 20 unit/μL, and the IVT reaction buffer 0.8x. The reaction was kept at 37 °C in a water bath for 3 h. DNase (1 unit/μL) was added to the reaction mixture and kept at 37 °C for 40 min to digest the extra plasmid template. For mRNA purification, LiCl solution (Sigma (St. Louis, MO, USA), L7026) was added at a final concentration of 2.5 M, and the mixture was kept at 4 °C for 30 min, allowing the precipitation of the IVT product. The precipitate was collected, washed with 70% ethanol, dried, resuspended with nuclease-free water, aliquoted, and stored at −80 °C until use.

### 2.3. Lipid Nanoparticle (LNP) Encapsulation

ALC-0315 and ALC-0151 were purchased from SINOPEG (Xiamen, China) Biotech Co., LTD., and 2-distearoyl-sn-glycero-3-phosphocholine (DSPC) and cholesterol were purchased from A.V.T. Pharmaceutical Co., Ltd. (Shanghai, China). The IVT product obtained from 3.2 was encapsulated in classic four-component LNPs (molar ratio of ALC-0315/ALC-0159/DSPC/cholesterol of 47.5/1.8/10/40.7) by PNI Nanoassemblr (Precision Nanosystems, Vancouver, BC, Canada). The water and ethanol flow rates were 9 mL/min and 3 mL/min, respectively. Assembled LNP samples were then ultra-filtered and stored at −80 °C with the addition of sucrose as a cryo-preservative.

### 2.4. Characterization and Quality Control

Both the mRNA product from 3.2 and the LNP product from 3.3 were subjected to standard QC procedures. The integrity was determined by a fragment analyzer 5200 (Agilent, Santa Clara, CA, USA), and products with >80% integrity were allowed for subsequent preclinical study. The size distribution was measured by Zetasizer Ultra Red Label (Malvern PANalytical (Malvern, UK)) with PDI < 0.2. The encapsulation efficiency (EE) was determined by the Quant-iT™ RiboGreen RNA kit (Thermo-Fisher (Waltham, MA, USA), R11490). The cryo-transmission electron microscope (cryo-TEM) graph was acquired by Krios G4 (internally named Einstein). The endotoxin amount was determined with the ToxinSensor™ Chromogenic LAL Endotoxin Assay Kit (GenScript (Nanjing, China)). The ratio of empty LNPs was detected using CYTOTM 9 staining reagent by the Nanoflow instrument. The ratio of mRNA to lipids and the mRNA copy number were determined by tangential flow filtration coupled with a multi-angle laser scattering instrument (TFF-MALLS (Gottingen, Germany)). The thermal stability of mRNA-LNPs was measured by qPCR. As the temperature increased, mRNA-LNPs dissociated and released free mRNA, binding to a fluorescent dye, SYBR gold, then the intensity was quantified by qPCR.

### 2.5. Cell Transfection

Human embryonic kidney cells (HEK293T) were seeded at a density of 3 × 10^5^ per well in a 24-well cell culture plate one day before the transfection experiment. mRNA-LNPs (CVG199 to CVG206, 0.5 µg) were added to the cells at 60%–70% confluency. After 24 h, both transfected cells and supernatant were collected.

### 2.6. Flow Cytometric Analysis

Cells were detached with TrypLE™ Express Enzyme (Gibco, 12604013, Waltham, MA, USA) and washed with PBS three times. The cells were then stained with VZV gE monoclonal antibody (9C8) (Santa Cruz, sc-56995, 1:500, Dallas, TX, USA) as the primary antibody and FITC-conjugated goat anti-mouse IgG as the secondary antibody (BD 554001, 1:50). Cells were analyzed by a CytoFLEX flow cytometer (Beckman Coulter, Brea, CA, USA).

### 2.7. Western Blot

The cells were collected and washed three times, then lysed with RIPA buffer (Millipore, 20-188, Burlington, MA, USA) on ice for 30 min. Cell supernatant was centrifuged at 12,000 rpm for 10 min to remove cell debris and concentrated with a 10 kDa ultrafiltration tube (Millipore, UFC901096).

Both cell lysate and cell supernatant samples were loaded and run on SDS-PAGE and transferred to PVDF membranes by eBlot L1 (GenScript). After 2 h of blocking in 5% skim milk (with the addition of 0.5% Tween-20), the membrane was washed with TBS-T three times and incubated with mouse anti-VZV gE antibody (Santa Cruz, sc-56995) or mouse anti-GAPDH antibody (Absin (Shanghai, China), abs137959) overnight at 4 °C. The membrane was washed with TBS-T three times and incubated with goat anti-mouse IgG HRP antibody (Abcam, ab150115, Cambridge, UK) at room temperature for 40 min. The membranes were visualized with Pierce™ ECL Western blot substrate (Thermo-Fisher Scientific, 34577, Waltham, MA, USA), and images were acquired with a GelView 6000 Plus (Guangzhou Biolight Biotechnology (Guangzhou, China)).

### 2.8. Animal Studies

Female SPF-graded C57BL/6j (6–8 weeks) mice, and Female SPF-graded SD (6–8 weeks) rats were purchased from Beijing Vitalstar Biotechnology Co. Ltd. and housed in individually ventilated cages (IVCs). All the animal experiments in this study were approved and supervised by the Animal Welfare Committee of CNBG-Virogin Co., Ltd. For the naïve mouse model, mice were immunized with mRNA-LNPs intramuscularly (IM) in the hind legs according to the immunization schedule depicted in Figure 1F(a,b). For the primed model, 10,000 pfu vOka, a live-attenuated virus product (LAV) purchased from Shanghai Institute of Biological Products Co., Ltd., was injected into mice subcutaneously 35 days before the study. Shingrix^®^, bought from GSK, was used as a benchmark in several studies. The subsequent immunization and blood collection schedule are depicted in Figure 1F(c). Blood was collected by puncture of the mandibular venous plexus, and sera were stored at −80 °C. At the endpoint, the mice were euthanized and their spleens harvested for cellular immune evaluation.

Rats were immunized with mRNA-LNP intramuscularly (IM) and intradermally (ID) in the hind legs, with an immunization interval of 14 days, as shown in Figure 1F(d). Blood was collected by puncture of the mandibular venous plexus, and sera were stored at −80 °C. PBMCs were harvested for cellular immune evaluation.Figure 1In vitro results for CVG-VZV mRNA vaccines and vaccination pattern schemes. All gE-expressing mRNA constructs evaluated in this study are shown (**A**). Gene apparatus such as signal peptide (SP), extracellular domain (ED), transmembrane domain (TM), ubiquitin tag (Ub), lysosomal-associated membrane protein 1 (LAMP1) tag, and linker are labeled. Mutations in CVG200, CVG203, and CVG204 are indicated. All CVG-VZV mRNAs were transcribed in vitro, and purified products were presented at the expected sizes that were loaded on an RNA gel (**B**). The encapsulated CVG-VZV mRNAs were transfected into HEK293T cells, and the in vitro expression of mRNA products that were above detectable levels was evaluated by gE protein antibody in flow cytometry and plotted in bars (**C**). The representative size distribution of CVG206 was determined using dynamic light-scattering analysis (**D**). Representative TEM characterization of CVG206 was performed, and the bleb formation in LNPs is marked by white arrows. A 50 nm scale bar is shown at the bottom-left corner (**E**). All the animal models, including C57BL/6j mice and SD rats, used in this study are shown (**F**), such as the naïve mouse model (*N* = 6) with two doses (5 μg of CVG-VZV/dose) of vaccination (**a**), the long-term (189 days) naïve mouse model (*N* = 6) with two doses (5 μg of CVG206/dose) of vaccination (**b**), the primed mouse model (*N* = 8) established by subcutaneous injection of 10,000 pfu vOka virus, then with two doses (either 5 μg or 20 μg of CVG206/dose) of vaccination after 35 days (**c**), and SD rat models (*N* = 5) for intradermal administration of CVG-VZV vaccine at various doses (**d**).
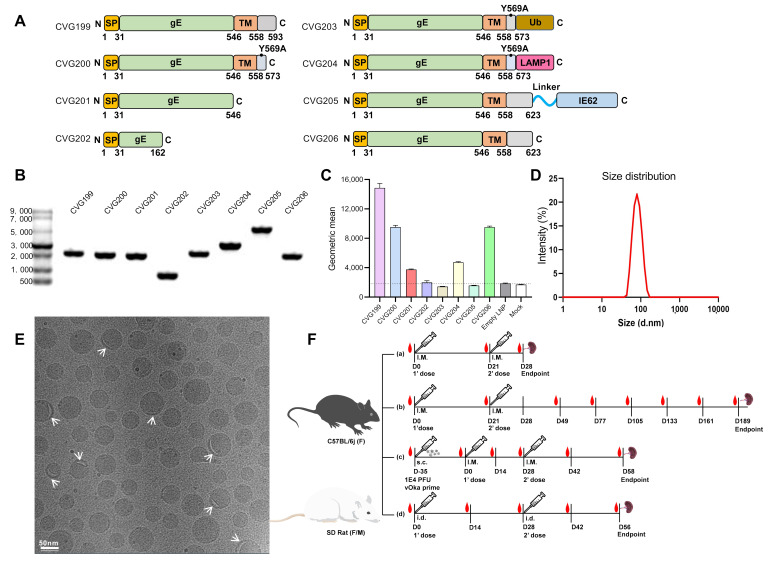


### 2.9. Enzyme-Linked Immunosorbent Assay (ELISA)

ELISA plates (Servicebio, 96-well ESP-96-D, Wuhan, China) were precoated with 100 ng gE protein (ACRO, GLE-V52H3-100 μg, Fareham, UK) per well. Serial dilution (starting with 1:10^3^ to 1:10^10^ in 3-fold dilution) was performed on the sera samples. The diluted sera samples were added to a gE-coated ELISA plate, incubated at 37 °C for 1 h, and washed thoroughly with TBST three times. Goat anti-mouse IgG H.R.P. antibody (Abcam, ab205719, 1:5000 dilution) was then added and incubated at 37 °C for 40 min. After triple washing with PBS–0.1% Tween solution, 50 µL T.M.B. substrate (Thermo Fisher Scientific, 34028) was added to each well and incubated at 37 °C for 20 min. Sulfuric acid (50 µL, 0.2 M) was added as a stopping solution, and plates were read at 450/620 nm by a SpectraMax iD3 (Molecular Devices, San Jose, CA, USA). The reciprocal antibody titers were determined if P/N > 2.

### 2.10. Intracellular Cytokine Staining (ICS)

Mouse spleens were homogenized with syringes and cell suspensions were filtered through a 70 μm cell strainer (BD FALCON, 352350, Franklin Lakes, NJ, USA) after red blood cell lysis (Beyotime, C3702-500 mL, Shanghai, China). The cells were resuspended with 1640 medium (Thermo Fisher Scientific, 61870036) supplemented with 10% fetal bovine serum (Gibco, 10099141C) and then were counted using a Countess™ 3 cell counter (Thermo Fisher Scientific). A total of 1 × 10^6^ cells were added to each well of a round-bottom 96-well plate and restimulated with 2 μg/mL peptides spanning gE protein (18 mers, 9 overlapping, synthesized by GenScript). A cell stimulation cocktail (Thermo Fisher Scientific, 00-4970-93) was used as a positive control. The 96-well plates were cultured in a humidified environment at 37 °C, 5% CO_2_ for 16–18 h. GolgiStop (B.D., 554715, freshly diluted in a 1:100 ratio) was added to each well and the culture was maintained for an additional 6 h. Afterward, cells were collected and stained with Fixable Viability Stain 510 dye (B.D., 564406). After blocking with rat anti-mouse CD16/CD32 Fc Block (B.D., 553141), cells were stained with a surface antibodies cocktail containing CD45 APC-CY 7 (B.D., 557659), CD3e PerCP-CY5.5 (B.D., 551163), N.K. 1.1BV650 (B.D., 564143), CD4 FITC (B.D., 557307), and CD8a AF700 (B.D., 557959). Cells were fixed with Cytofix (B.D., 554722) at 4 °C for 30 min and permeated with per wash buffer (BD 554723). Cells were incubated with IL-2 PE (B.D., 554428) and I.F.N.- A.P.C. (B.D., 554413) at 4 °C for 30 min to detect intracellular cytokine expression. Cells were washed and resuspended with staining buffer and analyzed by the CytoFLEX LX flow cytometer (Beckman Coulter). The results were analyzed using FlowJo_10.8.1 software and plotted using GraphPad Prism 9 software.

Rat PBMCs were collected after red blood cell lysing (BOSTER, AR1118, Pleasanton, CA, USA), resuspended with 1640 medium (Gibco, 61870036) supplemented with 10% fetal bovine serum (Gibco, 10099141C), and then counted using the Countess™ 3 cell counter (Thermo Fisher Scientific). A total of 1 × 10^6^ cells were added to each well of the round-bottom 96-well plate, then restimulated with 2 μg/mL peptides spanning gE protein (18 mers, 9 overlapping, synthesized by GenScript). The 96-well plates were cultured in a humidified environment at 37 °C, 5% CO_2_ for 16–18 h. The GolgiPlug (B.D., 555028, freshly diluted in a 1:1000 ratio) was added to each well, and the culture was maintained for an additional 6 h. Afterward, cells were collected and blocked with BSA (BBI, E661003-0100), then stained with a surface antibody cocktail containing Fixable Viability Stain 660 (FVS660) (B.D., 564405), BV421 anti-Rat CD3 (B.D., 563948), BV711 anti-Rat CD4 (B.D., 747023), and BV605 anti-Rat CD8 (B.D., 740370) at 4 °C for 30 min. Cells were fixed with Fixation Buffer (BioLegend, 420801) at room temperature for 20 min and washed with PBS (Adamas life, C8020). Cells were resuspended with PBS and placed at 4 °C for 16 h, then permeabilized with Intracellular Staining Perm Wash Buffer (BioLegend, 421002) and incubated with FITC Anti-Rat IFN-γ (B.D., 1230480) at 4 °C for 60 min for intracellular cytokine expression detection. Washing and resuspension were performed again before analysis by the flow cytometer (B.D.).

### 2.11. Enzyme-Linked Immunosorbent Spot Assay (ELISpot)

A total of 3 × 10^5^ cells of mice splenocytes were added to each well of IFN-γ ELISpot plates (Mabtech, 3321-4APT-10, Nacka Strand, Sweden) and IL-2 ELISpot plates (Mabtech, 3441-4APW-2), and 2 μg/mL peptide pool solution was added as a stimulant. The cells were kept in a humidified environment at 37 °C, 5% CO_2_ for 16–18 h. Following the manufacturer’s instructions, cells were discarded, and the 96-well plates were washed five times. Detection antibody was added and incubated at room temperature for 2 h. Plates were washed and streptavidin–ALP was added, with incubation at room temperature for 1 h. Plates were washed and prefiltered BCIP/NBT-plus solution was added for 5–20 min until dark spots were clearly visible, then rinsed with tap water, dried, and read by AID iSpot ELISpot FluoroSpot reader (Autoimmun Diagnostika GmbH, Straßberg, Germany) or IRIS FluoroSpot/ELISpot reader (Mabtech). The results were plotted using GraphPad Prism 9 software.

### 2.12. Mouse Inflammatory Cytokine Cytometric Bead Array (CBA) and Th1/Th2 Cytokine CBA

Mouse inflammatory cytokine analysis was performed on the serum samples according to the manufacturer’s instructions (BD, 552364). For Th1/Th2 cytokine analysis, 1 × 10^6^ mice splenocytes were plated into each well of a round-bottom 96-well plate and restimulated with 2 μg/mL 18-mer, 9-overlapping gE peptides. The plate was cultured in a humidified 37 °C, 5% CO_2_ incubator for 16–18 h, and the supernatant was collected and subjected to CBA analysis according to the manufacturer’s instructions (BD, 551287). FCAP array v3.0 was used to calculate the cytokine concentration.

### 2.13. RNA Optimization by AI Automation Screening Platform

Referring to the CVG206 amino acid sequence, ten different CDS sequences were designed by a self-developed AI algorithm. Xenopus β-globin UTR sequences were used on all 5′ and 3′ ends of the RNAs. A HiBiT tag was added to each RNA 3′ end for more accurate and sensitive expression evaluation at various time points (19 h, 24 h, and 48 h) on the automated experimental platform. The HiBiT tag was deleted after sequence optimization before any in vivo experiments. The AI-based experimental platform automatically executed the entire screening process, including in vitro transcription, cell transfection, and expression detection. Expi293F cells were used for RNA-LNP transfection. HiBiT detection is described in the Promega Nano-Glo^®^ HiBiT Lytic Detection System.

### 2.14. Lyophilization

The CVG206 mRNA was encapsulated using a jet-mixing approach and then proceeded to lyophilization. The lyophilization prescription of the lyophilized VZV mRNA vaccine consisted of 400 μg/mL CVG206 mRNA, sucrose, and a Tris system, with PEG-1500, sodium glutamate, and glycerol added as lyophilized protective agents. The encapsulation rate, integrity rate, particle size, zeta potential, particle distribution, mRNA polymer, moisture, mRNA content, insoluble particle, morphology, translation activity, and mRNA–lipid adduct of the nine-month study on lyophilized toxicology batch products at 2–8 °C produced in January 2024 are collected. All criteria were satisfied, revealing the excellent stability of the CVG206 lyophilized vaccine developed by our pilot-scale process techniques.

### 2.15. AlphaFold Prediction

For this study, we utilized AlphaFold version 2.3.2 to predict the 3D structure of the protein expressed by CVG206 mRNA. Following the open-source license, the original code, trained model weights, and inference script can all be obtained from the official GitHub database (https://github.com). AlphaFold 2 was deployed on a local workstation with an Nvidia RTX 4090 GPU (24 GB video memory), 120 GB RAM, and an Intel i9-13900K processor. During runtime, TensorFlow v.2.13.0, Python v.3.8.0, JAX v.0.4.10, and OpenMM v.8.0.0 were used. During the prediction process, the default parameter settings of AlphaFold 2 were followed, with “--db_preset = full_dbs and --use_gpu_relax = True” set to select the prediction result with the highest pLDDT. Structure visualization was created in PyMOL v.2.5.0.

### 2.16. Statistical Analysis

GraphPad Prism software 8.0 and 9.0 were used for the statistical analyses indicated in the figure legends. All statistical analyses were performed using the Kruskal–Wallis non-parametric test. A p-value less than 0.05 (* *p* < 0.05, ** *p* < 0.01, *** *p* < 0.001) was considered statistically significant.

## 3. Results

### 3.1. Rational Design and Characterization of Various VZV mRNA Constructs

The antigen VZV glycoprotein E protein was selected because of its significant abundance as an envelope protein with high immunogenicity. Full-length gE contains a signal peptide (SP, 1-31AA), extracellular domain (ED, 32-546AA), transmembrane domain (TM, 547-558AA), and C-terminus tail (C, 559-623AA). Different gE-based mRNA constructs were designed, either with full-length or truncated gE, and recombined with intracellular targeting tags or the transcriptional regulation protein IE62, respectively. A schematic representation of various mRNA constructs is presented in Figure 1A. CVG199 encodes a truncated 593 aa-length gE protein, starting from the N-terminus. CVG200 encodes a truncated 573 aa-length gE protein with a Y569A mutation [21]. CVG201 encodes a truncated 546 aa-length gE with a deletion of the transmembrane domain, which makes the gE protein secretable. CVG202 encodes the first 162 aa of the gE protein, which is enriched by B and T cell epitopes [33], and is intended to enhance the adaptive immune response by facilitating the presentation of antigen peptides to MHC-specific immune cells. CVG203 encodes a fusion protein consists of CVG200 expressed protein and a ubiquitin tag at the C-terminus. The ubiquitin tag was supposed to facilitate the presentation of fused antigens to CD8+ T cells by navigating encoded proteins to the ubiquitin–proteasome pathway [34,35]. CVG204 encodes a fusion protein that consists of CVG200 expressed protein and a lysosomal-associated membrane protein 1 (LAMP-1) tag at the C-terminus. LAMP-1 was expected to increase the trafficking of antigens to the endosome-associated compartment, thus enhancing the presentation of MHC II epitopes and facilitating the CD4+ T cell immune response [36,37]. CVG205 was designed as a fusion protein of the full-length gE and the immediate-early protein (IE62), which is a major transcriptional regulatory protein located in the nucleus in VZV-infected cells [38]. CVG206 encodes the full-length gE protein. All vaccine mRNA sequences were optimized based on the “population immune algorithm” with more than 200 factors involved in gene expression, including codon adaptation index (CAI), GC content, secondary structure, minimum free energy (MFE), etc. [39]. All 3D structures of expressed proteins by mRNA vaccines were predicted utilizing AlphaFold version. 2.3.2, and visualized in PyMOL v.2.5.0.

### 3.2. mRNA Vaccine Preparations: In Vitro Transcription (IVT), Lipid Nanoparticle (LNP) Encapsulation, Physicochemical Characterization, and In Vitro Expression

The eight VZV mRNA modalities were synthesized by the “one-pot” IVT method with capping and pseudo-uridine modification. Native gel (1% agarose) electrophoresis confirmed the expected size and purity of the IVT products (Figure 1B). mRNA integrity was measured by capillary electrophoresis. Mix-and-jet microfluidic technology encapsulated each IVT product into four-component lipid nanoparticles (LNPs). Although a difference existed in the mRNA cargo, all the mRNA-LNPs had similar size distribution patterns, with a mean size of 92 nm when characterized by dynamic light scattering (DLS) (Figure 1D). The mRNA-LNPs exhibited a polydispersity index (PDI) below 0.2, indicating a narrow size distribution characteristic of highly uniform nanoparticle populations. The LNPs exhibited either vesicular or solid electron-dense morphology when imaged using cryogenic transmission electron microscopy (cryo-TEM) (Figure 1E).

The expression levels of different mRNA constructs were determined by transfecting HEK293F cells (3E5 cells: 0.5 μg mRNA). All encapsulated CVG-VZV mRNAs were characterized by integrality, particle size, polydispersity (PDI), zeta potential, and encapsulation efficiency, and are listed in Table 1. The cell surface expression of gE was determined by flow cytometry after 24 h of transfection. Among the eight constructs, CVG199, CVG200, and CVG206 demonstrated much high surface expression levels, which were approximately fivefold or much higher than those of CVG202, CVG203, and CVG205 (Figure 1C). CVG202, CVG203, and CVG205 showed a low or negligible surface expression that was no different when compared to controls, empty LNPs, and mock. CVG201 showed slight significance compared to controls. Due to the failure to translate proteins in vitro of CVG202, CVG203, and CVG205, they were excluded from further study.

### 3.3. Behavioral Observation and Inflammatory Detection in Mice 18 h Post-Immunization

A mouse nesting behavior test was conducted to evaluate the general well-being of the animals after vaccination [40]. Briefly, a piece of pressed cotton (5 × 5 cm × 0.2 cm in size) was positioned in the center of each home cage of mice after immunization. Images were obtained 18 h after the first immunization (Figure 2A,D). As shown in Figure 2A,D, the cotton squares in the CVG200, CVG204, CVG206, LAV, and dPBS groups were torn apart to various degrees, resulting from normal nesting activity, whereas in CVG199-, CVG201-, empty LNP-, and Shingrix^®^-treated cages, the cotton square remained relatively intact. In the Shingrix^®^-treated cage, mice curled up in a corner with the cotton square untouched. The animals showed significantly reduced locomotor activity and nesting behavior, suggesting severe adverse reactions after vaccination.

We also measured the water, food consumption, and weight loss 18 h after the first immunization (Figure 2B,E). Slightly decreased water and food intake was observed in all CVG-immunized groups of mice compared with the dPBS control group. Mice vaccinated with CVG201 showed the lowest food consumption, which was consistent with the loss of nesting behavior in the group. The weight loss of CVG206-immunized mice showed a dose-dependent manner. CVG206 5 μg caused a 4% to 6% weight loss compared to 9% caused by 10 μg and 20 μg CVG206 immunizations. However, Shingrix^®^-treated and empty LNP-treated groups showed severe weight loss that was more than 11%, indicating potential discomfort and adverse effects from inoculations (Figure 2E). Although LAV exhibited good feedback to immunization, it revealed a much lower immunogenicity based on subsequent data.

We also investigated systemic inflammation reactions in vaccinated animals. Sera were collected 18 h after the first immunization, and the inflammation detection panel was assessed on all mouse serum samples by cytometric bead analysis. The results indicated that acute proinflammatory cytokine IL-6 and chemokine MCP-1 (monocyte chemotactic protein 1) expression in the serum samples were upregulated by vaccination. CVG199, CVG201, and CVG206 elicited higher IL-6 expression than CVG200, CVG204, and empty LNP groups (Figure 2C).

### 3.4. Robust Immune Responses Induced by VZV mRNAs in the Naïve Mouse Model

Naïve mice were immunized with VZV mRNA vaccines according to the schedule shown in Figure 1F(a). After euthanasia and sample collection, we performed a humoral immune response assessment of the serum samples on days 21 and 28. The gE-specific IgG antibody levels are shown in Figure 3A, in which CVG199, CVG200, CVG201, CVG204, and CVG206 elicited significant numbers of antibodies. We also tried CVG202, CVG203, and CVG205 mRNAs, although they showed unsatisfying low expression in our previous study. They indeed induced very low antibody levels, as expected, which are not shown here. The empty LNP and dPBS groups served as negative controls, and no gE-specific IgG antibodies were detected.

Cellular immune responses induced by VZV mRNAs were detected using three different approaches. Mouse splenocytes were harvested and restimulated with gE-overlapping peptides that were 18 amino acids in length with a 9 AA overlap, and then seeded in 96-well plates for later cytokines, such as IL-2 and IFN-γ detection by ELISpot or ICS. CVG199 and CVG206 showed more spot-forming units (SFU) stimulated per 3E5 splenocytes than the other groups, as shown in Figure 3B. The percentage of IL-2- and IFN-γ-expressing CD4+ or CD8+ T cells was also measured by flow cytometry. CVG206 mRNA generally induced more CD4+ and CD8+ T cells that expressed IL-2 and IFN-γ than other vaccine candidates, demonstrating higher immunogenicity (Figure 3C). Interestingly, compared with CVG206, all the other CVGs encoding truncated gE proteins led to lower stimulation of cellular immune response, suggesting that deletions of gE protein carboxyl end, especially the transmembrane domain deletion, may compromise vaccine efficacy.

In mice, lymphocytes of the Th1 subset are characterized by their ability to secrete IL2 and IFN-*γ*, which also promotes B cell secretion of IgG2a and inhibits IgG1. A low IgG2a/IgG1 ratio indicates a Th2-type humoral immune response, and a higher ratio indicates a Th1-type cellular immune response [41,42]. Higher IgG2a is always associated with strong cell-mediated cytotoxicity (ADCC) and opsonophagocytic effects by macrophages. [43]. To elucidate the status of the balance of systemic Th1/Th2-type responses, we assessed the IgG subclasses IgG1, IgG2a, IgG2b induced by CVG-VZV mRNAs in mice sera by ELISA. The IgG2a/IgG1 ratios of CVG199, CVG200, CVG201, CVG204, and CVG206 were higher than 1, indicating that CVG-VZV mRNA induces a Th1-skewed immune response (Figure 3D).

**Figure 3 vaccines-13-00327-f003:**
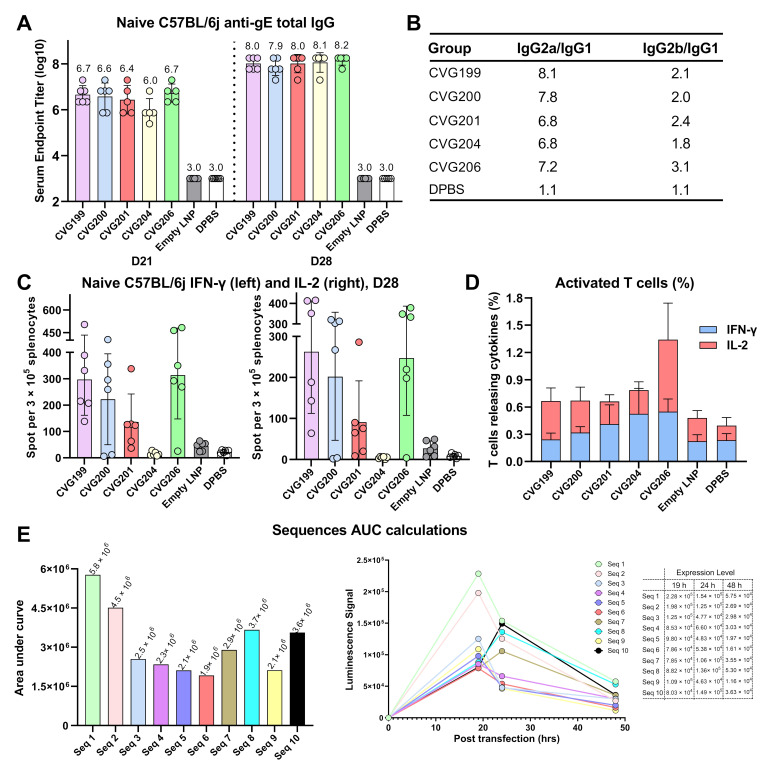
Humoral and cellular responses of CVG-VZV LNP in a naïve mouse model and CVG206 mRNA sequence optimization using an automated screening platform. Detection of VZV gE-specific IgG of all CVG groups at days 21 and 28 (**A**). Titers of IgG subtypes were measured by ELISA, and the ratios of IgG2a/IgG1 and IgG2b/IgG1 calculated for each group are shown (**B**). C57BL/6j splenocytes were restimulated with a gE-overlapping peptide pool, and the spot-forming units (SFUs) of T cells secreting IFN-γ (left) or IL-2 (right) were quantified using an ELISpot assay (**C**). The percentage of activated T cells (CD4+ and CD8+) releasing IFN-γ and IL-2 was measured by flow cytometry (**D**). Ten mRNA sequences of the CDS region were synthesized based on the same amino acid sequence of CVG206. The 5′ and 3′ UTRs of Xenopus β-globin were used in all the candidates. All ten mRNA candidates were encapsulated and transfected into HEK293F cells. The expression levels of all candidates were measured at 19 h, 24 h, and 48 h, then plotted on the same graph ((**E**), right). The area under the curve (AUC) of all candidates was calculated, indicating the approximate total expression level ((**E**), left).

### 3.5. mRNA Sequence Optimization of CVG206 on Protein Accumulated Expression by Automated Screening Platform

We selected CVG206 coding full-length gE as the candidate for further study due to its satisfying immunogenicity evaluated on the naïve animal model. We then optimized the mRNA sequences of the CDSs and UTRs. By adjusting and introducing codons with an appropriate codon adaptive index (CAI), calculating the minimum free energy (MFE) of RNA secondary structures, and combining with appropriate UTRs, the expression, stability, and half-life of the mRNA were significantly improved. We built an AI-oriented automated screening platform by collecting the expression values of numerous mRNA sequences from wet lab experiments to feed AI and develop an algorithm. Ten CVG206 mRNA sequences were obtained using optimized CDSs and UTRs. The expression levels of all CVG206 mRNAs were quantified at various time points (Figure 3E, right), and the total protein expression of each sequence was calculated using AUC accumulation (Figure 3E, left). Seq 1 exhibited 5.5 million units, which was threefold higher than that of Seq 6 (lowest expression), indicating significant optimization. Seq 1 was used as the optimized mRNA sequence of CVG206 for all subsequent studies.

### 3.6. 3D Structure Predictions and Immunogenicity Distribution Analysis of Full-Length and Derived Truncations of gE Protein

To explore the most immunogenic region of the gE protein and explain the attenuation caused by truncations, we analyzed the immunogenicity contributions of various portions of the gE protein (Figure 4). The number of IL-2 and IFN-γ spots induced by CVG206, which expressed the full-length gE protein (1 to 623 aa), was considered a 100% immunogenicity contribution. CVG201, which expressed a protein chunk from aa 1 to aa 546, contributed 30.8% of total IL-2 spots and 36.5% of total IFN-γ spots (in gray, Figure 4). CVG200, which expressed the truncated gE protein (from 1 to 573 aa), induced almost two-thirds of IL-2/IFN-γ immune spots: 68.3% of total IL-2 spots and 59.6% of IFN-γ spots (in gray and blue). CVG199, which expressed truncated gE protein (from 1 to 593 aa), induced 88.8% of total IL-2 spots and 80% of total IFN-γ spots (in all colors but green). Accordingly, various protein chunks contributed differently, as shown in Figure 4B. For the relatively smaller portions from aa 546 to aa 573, aa 573 to aa 593, and aa 593 to aa 623, they contributed equal or more immunogenicity than the major portion from aa 1 to aa 546. Therefore, it seems that the C-terminal ~70 aa (~11%) of the total length of the gE protein contributes more than 60% of the cellular immunity.

**Figure 4 vaccines-13-00327-f004:**
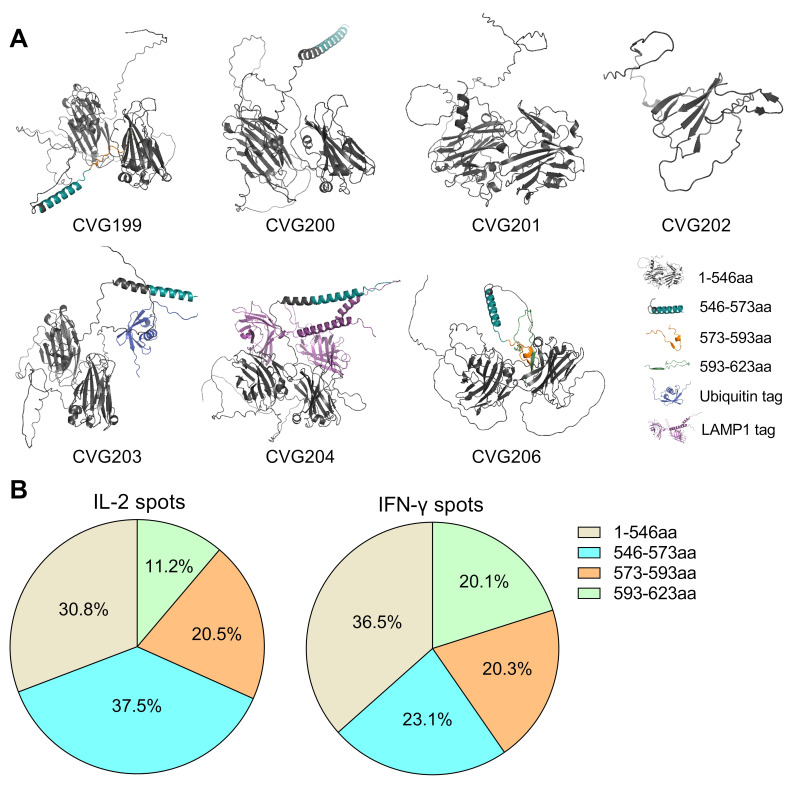
The predicted structures of truncated gE protein and immunogenicity distribution analyses of all CVG-VZV mRNA vaccines. The predicted structures of truncated gE protein expressed by CVG-VZV mRNAs were simulated using AlphaFold version 2.3.2 and presented in PyMOL version 2.5.0 (**A**). CVG199 to CVG202 expressed various truncated gE proteins (1 to 593 aa, 1 to 573 aa, 1 to 546 aa, and 1 to 162 aa); CVG203 and CVG204 expressed truncated gE (1 to 573 aa) modified with a functional tag. CVG206 expressed a full-length gE protein with no modification. Various chunks of protein were divided and are marked in different colors according to truncation size. The immunogenicity contributions by protein chunks were calculated and plotted (**B**), elucidating the various inducible abilities of immune responses by different portions of the gE protein.

### 3.7. Dose Dependence of Humoral and Cellular Immune Responses to the CVG206 Vaccine in Mouse Model

CVG206 was selected for further study due to its promising induction of humoral and cellular immune responses and acceptable side effects. Four doses of CVG206 (2.5 μg, 5 μg, 10 μg, and 20 μg) were administered to naïve C57BL/6j female mice in a two-dose regimen (Figure 1F(a)). vOKa, a live-attenuated virus (LAV), and Shingrix^®^ served as positive controls. The effectiveness of CVG206 was determined by levels of gE-positive IgG antibody (detected by ELISA) and cytokine expression (detected by ELISpot, flow cytometry, and CBA). The gE-specific IgG titers of the CVG206 group increased in a dose–response manner. A 5 μg dose of the CVG206 vaccine induced a comparable humoral immune response to LAV (1000 pfu vOKa/dose) and Shingrix^®^ (5 μg protein/dose). The 10 μg and 20 μg doses of CVG206 stimulated much higher IgG levels, measured on post-vaccination days 21 and 35 (Figure 5A). Splenocytes collected on day 35 from different dose groups were restimulated with the gE-specific peptide pool before being evaluated by ELISpot, ICS, and CBA. The expression levels of IFN-γ, TNF-α, and IL-2 induced by CVG206, especially in the 20 μg group, were higher compared to the LAV and Shingrix^®^ groups (Figure 5B–D), elucidating the more robust immune responses induced by CVG206. Overall, compared to the commercial live-attenuated vaccine (LAV) and subunit vaccine Shingrix^®^, the 20 μg dose of the CVG206 mRNA vaccine demonstrated its superiority in terms of immunoreactivity.

### 3.8. Long-Term Immunogenicity Evaluation of CVG206

The long-term immunogenicity induced by the CVG206 vaccine was evaluated on C57BL/6j female mice (Figure 6) following the inoculation scheme shown in Figure 1F(b). Four doses (5 μg, 10 μg, 20 μg, and 40 μg) of CVG206 were administered to the mice, with Shingrix^®^ (5 μg protein/dose) serving as a positive control. The gE-specific IgG titer was monitored over a period of six months. Samples were collected at seven time points, as shown in Figure 6A. The 5 μg and 10 μg CVG206 groups exhibited the same level of anti-gE IgG production as Shingrix^®^ throughout the evaluation period. The 20 μg dose of CVG206 induced a higher anti-gE IgG titer than Shingrix^®^ for the first 105 days, then dropped to the same level as Shingrix^®^. The 40 μg dose of CVG206 maintained a stronger long-term immunogenicity trend than the Shingrix^®^ group until the end of the experiments. Splenocytes from all groups were isolated and restimulated with the gE peptide pool on day 189 for cellular immune response evaluation. The SFU of IFN-γ and IL-2 were measured by ELISpot (Figure 6B), long-term CD4+ and CD8+ T cell counts releasing related cytokines were measured by flow cytometry (Figure 6C), and the secretions of IFN-γ and IL-2 in splenocyte supernatants were detected using a CBA kit (Figure 6D). Compared to 5 μg Shingrix^®^, all doses of CVG206 showed a more robust cellular immune response.

### 3.9. Immunogenicity Induced by CVG206 in a VZV-Primed Animal Model

Since there is no appropriate VZV-reactivation animal model, assessing the VZV vaccine’s efficacy in preventing the reactivation of latent VZV is difficult. A VZV-primed animal model (Figure 1F(c)) was used to simulate VZV infection in the natural world, as described in previous studies. Briefly, 10,000 pfu vOKa was administered to the mice before immunization to build up the seropositive model. On day 35 post-virus inoculation, seropositivity was confirmed by detecting gE-specific IgG. On the same day, the vOKa-primed C57BL/6j female mice received 5 μg and 20 μg of lyophilized CVG206 (Table 2), LAV or 5 μg Shingrix^®^. Animals in the control groups (primed only) were also primed with the same vOKa virus, followed by receiving either empty LNPs or dPBS. Samples from several time points were collected to plot the IgG titer curves, as shown in Figure 7A. The IgG titers across all immunized groups increased on day 28 and continued to rise throughout the experiment, following the trend: vOKa < Shingrix^®^ ≤ 5 μg CVG206 < 20 μg CVG206. The splenocytes of all mice were harvested on day 58 for further immunogenicity evaluations. Interestingly, the LAV group, primed with 10,000 pfu and boosted with 2000 pfu vOKa, showed considerable and sustained increasing expression of gE-specific IgG compared to the dPBS control. Thus, both CVG206 and Shingrix^®^ can induce robust humoral and cellular immune responses in vOKa-primed animals. CVG206 at 5 μg and 20 μg performed equal to or better than 5 μg Shingrix^®^ in all evaluations (Figure 7B–D).

**Figure 7 vaccines-13-00327-f007:**
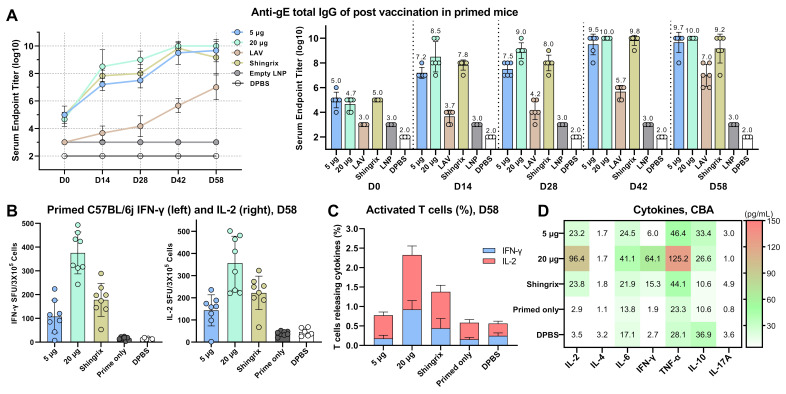
The humoral and cellular responses induced by lyophilized CVG206 mRNA vaccination in the primed animal model. As described in Figure 1F, the VZV primed mouse model was established by subcutaneous injection of 10,000 pfu vOka virus into C57BL/6j naïve mice. Higher gE-specific total IgG was induced by Shingrix^®^ as well as 5 μg and 20 μg of CVG206 at various time points. Live-attenuated virus (LAV), vOKa, and Shingrix^®^ served as positive controls, while empty LNPs and dPBS served as negative controls (**A**). The IgG levels are presented in curves (left) and value bars (right). VZV-primed C57BL/6j mice splenocytes (D58) were restimulated with a gE-overlapping peptide pool, and the spot-forming units (SFUs) of T cells secreting IFN-γ (left) or IL-2 (right) were quantified using an ELISpot assay (**B**). The percentages (D58) of activated T cells releasing IFN-γ and IL-2 in all groups were measured by flow cytometry (**C**). Cytokines released by splenocytes isolated from immunized mice (D58) from all groups after stimulation with the gE-overlapping peptide pool were measured by cytometric bead assays (CBAs) (**D**).

**Table 2 vaccines-13-00327-t002:** The lyophilized VZV mRNA vaccine, CVG206 thermal stability study. The pilot-scale VZV mRNA vaccine CVG206 was manufactured in January 2024 as patch 1, so-called 202401001. The 202401001 samples were lyophilized at CNBG-Virogin facility and proceeded to thermal stability study. The encapsulation, integrity, size, zeta potential, particle distribution, etc., were measured or observed on various monthly time points. All of the criteria are listed in the table as quality controls.

Patch	Temp.	Month	Encapsulation (%)	Integrity (%)	Size (nm)	Zeta Pot. (mV)	Particle Distribution	mRNA Polymer (%)
202401001	2–8 °C	0	92.6	84.3	78	2.0	Dv (10):42 Dv (50): 61 Dv (90):100	9.5
	2–8 °C	1	90.6	83.6	78	0.5	Dv (10):45 Dv (50): 64 Dv (90):102	10.6
	2–8 °C	2	91.0	83.2	77	1.0	Dv (10):40 Dv (50): 58 Dv (90):98	9.1
	2–8 °C	3	92.4	81.8	76	2.0	Dv (10):40 Dv (50): 59 Dv (90):96	9.8
	2–8 °C	6	89.6	82.6	75	2.0	Dv (10):44 Dv (50): 63 Dv (90):96	11.9
	2–8 °C	9	89.9	81.7	75	2.0	Dv (10):40 Dv (50): 58 Dv (90):94	12.7
	2–8 °C	criterion	≥80.0	≥65.0	50~150	−30~30		≤15
Patch	Temp	Month	Moisture (%)	mRNA Content (µg)	Insoluble Particles	Morphology	Translation Activity	mRNA–Lipid Adducts (%)
202401001	2–8 °C	0	0.5	56	≥10 µm: 792 ≥25 µm: 7	White, loose	High	0.6
	2–8 °C	1	1.0	51	≥10 µm: 1914 ≥25 µm: 13	White, loose	High	0.6
	2–8 °C	2	1.0	48	≥10 µm: 2144 ≥25 µm: 11	White, loose	High	0.6
	2–8 °C	3	1.1	49	≥10 µm: 1750 ≥25 µm: 14	White, loose	High	0.6
	2–8 °C	6	0.9	49	≥10 µm: 2133 ≥25 µm: 19	White, loose	High	0.7
	2–8 °C	9	0.9	56	≥10 µm: 2447 ≥25 µm: 19	White, loose	High	1.8
	2–8 °C	criterion	≤3	40~60/Vial	≥10 µm: 6000 ≥25 µm: 600	White, loose	≥50%	≤10

### 3.10. Intradermal Administration of CVG206 mRNA Vaccine Induced Comparable Robust Immune Responses at Much Lower Doses

We further investigated the immune response to CVG206 via different routes of vaccination, i.e., intramuscular (IM) and intradermal (ID). CVG206 was administered intradermally and intramuscularly at various doses for comparison in rats. As we are under the VZV mRNA vaccine IND application process, the clinical doses we will use in clinical trials are confidential and cannot be disclosed. For further discussion, “dose” is used as a vaccination unit for description. For the intramuscular groups, IM “1/3 dose” μg/150 μL and IM “1 dose” μg/500 μL were vaccinated at the same concentration; All intradermal groups, ID “1/12 dose” μg, ID “1/6 dose” μg, and ID “1/3 dose” μg, were administered at the same injection volume, 150 μL, since the larger the volume, the more Langerhans cells that are DC-like macrophages that elicit antigen presenting would be contacted. The vaccination of different volumes will lead to different antigen-presenting efficiency and vaccine-induced immunogenicity; thus, it is crucial to maintain a specific and manageable volume of inoculation. The anti-gE IgG antibody levels in sera were quantified at D14, D28, D42, and D56 (Figure 8A). The ID “1/6 dose” μg of CVG206 induced a comparable anti-gE specific IgG response to IM “1/3 dose” μg of CVG206, and ID “1/3 dose” μg of CVG206 induced a comparable anti-gE specific IgG response to IM “1 dose” μg of CVG206. Interestingly, ID “1/12 dose” μg of CVG206 even induced a decent amount of the anti-gE specific IgG level, comparable to the IM “1/3 dose” μg of CVG206 at D56 (Figure 8A-right). The ID “1/6 dose” μg and “1/3 dose” μg of CVG206 elicited more robust cellular immune responses than the IM “1/3 dose” μg and “1 dose” μg of CVG206 in the rat model, as measured by ELISpot (Figure 8B) and flow cytometry CD4+ and CD8+ IFN-γ release (Figure 8C).

## 4. Discussion

In this study, the integration of AlphaFold (version 2.3.2) prediction and in vivo ELISpot evaluation in a naïve mouse model provided critical insights into the structural basis of immunogenicity variations observed across truncated forms (CVG199 to CVG206) of varicella-zoster virus (VZV) glycoprotein E (gE) (shown in Figure 4). By predicting the 3D structures of gE truncations (e.g., CVG199: 1–593 aa, CVG200: 1–573 aa, CVG201: 1–546 aa) and immunospot counting, AlphaFold explained the potential of a systematic correlation between structural domains and their functional contributions to cellular immunity. The study revealed that the C-terminal region (~546–623 aa), despite constituting only ~11% of the full-length gE protein, harbors structurally dynamic motifs that likely enhance antigen presentation and T cell epitope exposure. This observation aligns with the experimental data showing that the C-terminal truncations (e.g., CVG201: 1–546 aa) contributed disproportionately lower IL-2/IFN-γ responses (30.8% and 36.5%, respectively) compared to constructs retaining the C-terminal region (e.g., CVG199: 1–593 aa induced 88.8% IL-2 and 80% IFN-γ responses). Furthermore, AlphaFold’s structural predictions facilitated the identification of potential conformational changes caused by truncations. For instance, the removal of residues 594–623 aa (as in CVG199) may disrupt stabilizing loops or solvent-exposed helices critical for MHC class II interactions, explaining the attenuated immunogenicity. Regarding all the structural prediction, immunogenicity distribution analysis and immuno-evaluations in naïve mice, our CVG206 VZV mRNA vaccine exhibited the most robust humoral and cellular immune responses and a noticeable advantage in terms of safety over all other inoculates. The expression of MCP-1 and IL-6 was significantly induced by CVG206 immunization (Figure 2C), which was caused by the specificity of the mRNA–LNP complexes, consequently inducing a native immune response. IL-6 can directly instruct the hypothalamus to increase body temperature, which can further prevent subsequent infections. MCP-1, or CCL2, promotes the recruitment and function of neutrophils and monocytes to facilitate pathogen clearance. Vaccination with CVG206 increased the percentage of VZV antigen-specific CD8+ T cells that released IFN-γ, which is crucial for cellular immunity against VZV.

As a top concern, the safety of the VZV mRNA vaccine was evaluated using multiple parameters. The “nestlet shredding” [40] behavior (Figure 2A) was observed in all CVG-VZV candidates and various doses of CVG206. The primary goal of the nestlet shredding experiment was to semi-quantitatively observe general pain and discomfort in animals caused by vaccination. Since nestlet shredding behavior is regulated by psychological, neurobiological, and physical factors, disruption of this behavior may reflect more comprehensive adverse effects of the vaccine [40,44]. Interestingly, mRNA vaccines at all doses showed less disturbance of nestlet shredding behavior than 5 μg Shingrix^®^, suggesting that the former may have less severe side effects. On the other hand, CVG206 showed more water and food consumption than the other mRNA vaccines, indicating its better safety potential.

Robust cellular immune response is crucial for the immune-protective effect of VZV vaccine against reactivation of latent VZV infection [15,30]. In the present study, mRNA constructs encoding various lengths of truncated gE protein were tested for their immunogenicity. CVG202, 203, and 205 were excluded from further studies due to very low protein production in vitro. Interestingly, while various truncated gE constructs induced comparable levels of antibodies against the VZV antigen, their cellular immunity differed significantly (Figure 3). Based on these results, we further estimated the contributions of different parts of the gE antigen to cellular immunity against VZV (Figure 4B). Our results revealed that epitopes for cellular immunity seem to concentrate in the C-terminal 70 aa. However, this finding needs further verification with epitope mapping, as truncation of the full-length gE may change levels of protein expression or patterns of proteolysis, which can alter the MHC processing of antigen presentation [45,46,47,48,49].

Based on numerous studies, intradermal administration is a more sophisticated approach for pursuing better efficacy and safety of vaccination. Unlike antigen delivery into muscle and vessels by intramuscular administration, intradermal immunization represents an alternative route with some advantages [50]. Langerhans cells, a group of DC-like macrophages, are located and enriched in the dermis [51], allowing for more efficient antigen presentation. Our results of intradermally delivered CVG206 showed a more robust cellular immune response with lower doses compared to intramuscular inoculation. Compared to IM, one-twelfth to one-third of the dose delivery achieves sufficient efficacy. Intradermal administration enables a dose-sparing strategy during shortages and reduces costs [52,53]. Additionally, many medical apparatuses for intradermal administration have been developed, such as needleless syringes, microneedle syringes, and dissolvable patches. These are all painless administrations, making them more acceptable in clinical settings [54]. The primary disadvantage is that the skin can only hold a limited volume of inoculates, which hinges on determining the optimal dose [55]; thus, the optimal ID volume will be investigated in our future study for pursuing the best vaccine efficacy in terms of the certain density of Langerhans and dermal dendritic cells (DDC) in the dermis. Furthermore, more immunological fundamental mechanisms of antigen-presenting immune cells recruiting and migrating related to draining lymph nodes stimulated by intradermal inoculation will be addressed as well. Other potential issues of intradermal delivery include the intensive technical training needed for administration [56], fluid waste issues due to leakage or dead volume [57], and potential side effects such as pain, swelling, and itching at the injection sites, which should all be tackled [58].

Recently, the approval of the respiratory syncytial virus (RSV) vaccine mRSVIA^®^ [59], the world’s first non-COVID mRNA vaccine, furnished confidence to both academia and industry in developing mRNA technology, although for young children, enhanced respiratory disease caused by vaccination (ERD) was observed. For adults, the RSV mRNA vaccine is indeed enormously effective in disease prevention. In our study, besides RNA sequence optimization by the AI-based screening platform, lyophilized LNP formulation and related manufacturing processes were developed to conquer thermal instability using the quality-by-design (QbD) approach. The physicochemical properties of the lyophilized formulation remained unchanged at 2–8 °C over 9 months, eliciting strong humoral and cellular immune responses in mice, similar to the liquid formulation. This contributes to a more achievable VZV vaccine in terms of cheaper and more straightforward storage and transportation for the LNP vaccine, making it more accessible to potential vaccinees. The intradermal administration of the CVG206 vaccine also yielded satisfying results, although more investigations on intradermal applications, both in terms of mechanism and inoculating apparatus, are needed.

## 5. Conclusions

In the post-SARS-CoV-2 pandemic landscape, mRNA vaccines have emerged as a transformative technology for addressing emerging infectious diseases, leveraging their proven advantages in rapid development and adaptability. By bridging structural predictions with functional immunogenicity assays, this study underscores AlphaFold’s and AI algorithm designs’ transformative role in accelerating antigen optimization for mRNA vaccines. The computational insights not only validated empirical findings but also provided a roadmap for future engineering of antigen-based vaccines to maximize cellular immunity against specific pathogen infections. Nevertheless, critical challenges persist, necessitating focused research to optimize lipid nanoparticle (LNP) delivery systems for tissue-specific targeting and enhanced safety profiles, establishing predictive humanized and pathogen-responsive animal models, resolving RNA design complexities arising from codon degeneracy and untranslated region (UTR) optimization through advanced AI-driven platforms, and standardizing quality control protocols for investigational new drug (IND) applications.

The CVG206 mRNA vaccine exemplifies next-generation innovation through its development pipeline: designed via a high-throughput automated screening platform powered by proprietary AI algorithms, produced using novel lyophilization techniques to ensure stability, and administered via intradermal delivery to enhance immunogenicity. These synergistic advancements position CVG206 as a thermally stable vaccine candidate with an optimized safety–efficacy balance. Preclinical data suggest its potential not only to prevent primary VZV infection but also to mitigate viral reactivation and associated neuropathic pain, potentially offering a groundbreaking therapeutic strategy for herpes zoster complications.

## Figures and Tables

**Figure 2 vaccines-13-00327-f002:**
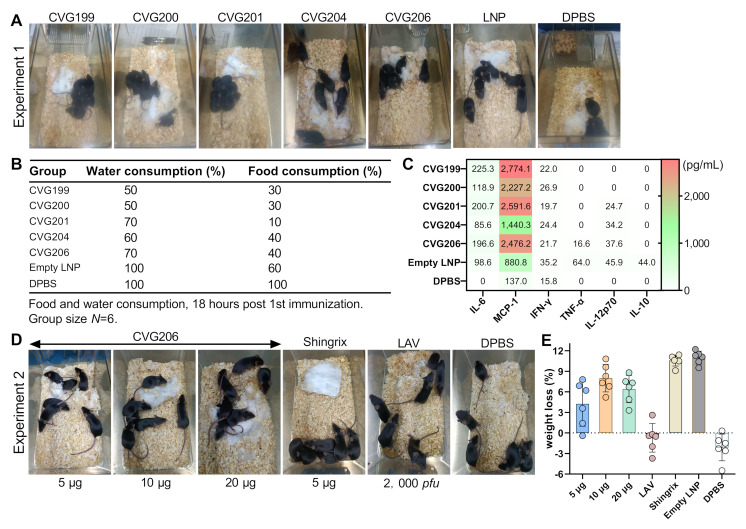
Side effect evaluations of mice vaccinated with 1st dose of CVG-VZV vaccine after 18 h. Nesting behavior was observed in all mice that were vaccinated with 1st dose after 18 h. Pressed cotton squares (5 × 5 cm) were positioned at the same location on one side of the cage. More substantial side effects after vaccination would remain in intact cotton because of the decrease in the action of mice. Active mice tear the cotton thoroughly, which indicates a more comfortable condition and fewer side effects. CVG-VZV mRNA vaccine-immunized mice served as the experimental groups, and empty LNP- and dPBS-treated groups served as negative controls (**A**). Water and food consumption were recorded to determine the condition of all groups (*N* = 6) (**B**). The expression of cytokines responsible for inflammation was quantified in all mice 18 h after the 1st vaccination (**C**). The various doses of CVG206 mRNA-, LAV-, and Shingrix^®^-vaccinated mice served as experimental groups, and the dPBS-treated group served as negative controls (**D**). The body weight loss (%) of various doses of CVG206 mRNA-, LAV-, and Shingrix^®^-vaccinated mice were measured 18 h after the first vaccination (*N* = 6), and the empty LNP- and dPBS-treated group served as negative controls (**E**).

**Figure 5 vaccines-13-00327-f005:**
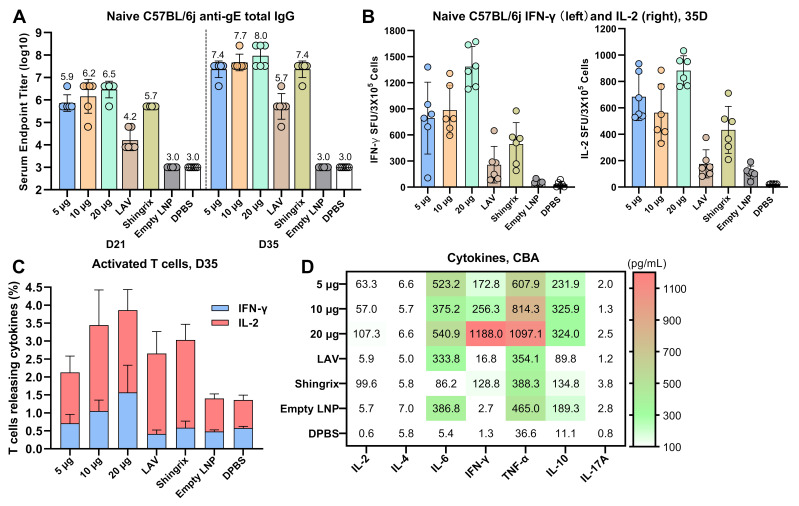
Humoral and cellular response induced by different doses of CVG206. The detection of gE-specific total IgG (D21, D35) induced by different doses of CVG206 and all controls. Live-attenuated virus, vOKa, purchased from Shanghai Institute of Biological Products Co., Ltd., and Shingrix^®^, purchased from GSK, served as positive controls; empty LNPs and dPBS served as negative controls (**A**). C57BL/6j splenocytes (D35) were restimulated with a gE-overlapping peptide pool, and the spot-forming units (SFUs) of T cells secreting IFN-γ (left) or IL-2 (right) were quantified using the ELISpot assay (**B**). The percentages (D35) of activated T cells (CD4+ and CD8+) releasing IFN-γ and IL-2 in all groups were measured by flow cytometry (**C**). Cytokines released by splenocytes isolated from immunized mice (D35) from all groups after stimulation with the gE-overlapping peptide pool were measured by cytometric bead assays (CBAs) (**D**).

**Figure 6 vaccines-13-00327-f006:**
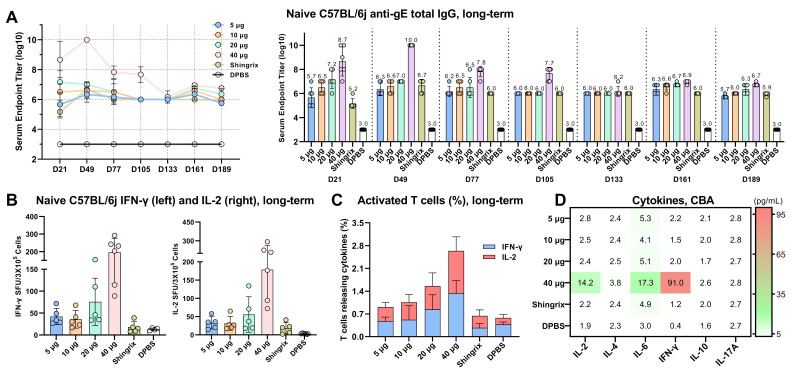
The long-term humoral and cellular responses induced by CVG206 of different doses. The gE-specific total IgG induced by different doses of CVG206 and other controls was detected at various time points over 6 months. Live-attenuated virus, vOKa, bought from Shanghai Institute of Biological Product Co., Ltd., and Shingrix^®^, bought from GSK, served as positive controls, and empty LNPs and dPBS served as negative controls (**A**). The anti-gE IgG levels are presented in curves (left) and value bars (right). C57BL/6j splenocytes (D189) were restimulated with gE-overlapping peptide pool, and the spot-forming units (SFUs) of T cells secreting IFN-γ (left) or IL-2 (right) were quantified with an ELISpot assay (**B**). The percentages (D189) of activated T cells (CD4+CD44+ and CD8+CD44+) releasing IFN-γ and IL-2 of all groups were measured by flow cytometry (**C**). Cytokines released by splenocytes isolated from immunized mice (D189) from all groups after stimulation with gE-overlapping peptide pool were measured by cytometric bead assays (CBAs) (**D**).

**Figure 8 vaccines-13-00327-f008:**
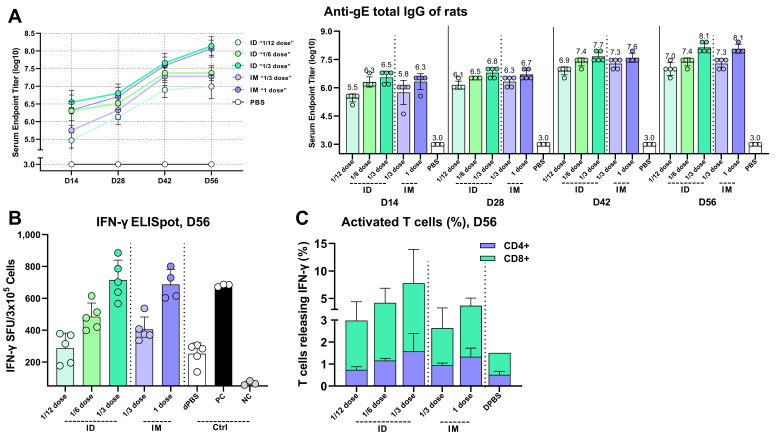
The intradermal administration of CVG206 induced robust immunogenicity at much lower doses in a naïve rat model. The intradermal study was performed on a rat model, as shown in {{float-placeholder-vaccines-13-00327-f001}}F(d). The “1/12 dose” μg, “1/6 dose” μg, and “1/3 dose” μg of CVG206 were administered to rats by intradermal (ID) injection. The “1/3 dose” μg and “1 dose” μg of CVG206 were administered to rats by intramuscular (IM) injection. The anti-gE specific IgG antibody levels were quantified in serum samples collected from the rats at time points D14, D28, D42, and D56 (**A**). The IFN-γ spot-forming units (SFUs) of restimulated splenocytes isolated on D56 were measured by ELISpot (**B**). The ratios of CD4+ and CD8+ T cells that released IFN-γ collected from restimulated splenocytes on D56 were measured by flow cytometry in rats (**C**). The dPBS group served as a negative control.

**Table 1 vaccines-13-00327-t001:** Physicochemical characterizations of encapsulated CVG-VZV mRNAs. All CVG-VZV mRNA solutions were encapsulated in a lipid nanoparticle formulation licensed by Acuitas. Quality inspections were performed on all the encapsulated CVG-VZV mRNA-LNP samples. Integrality was measured by capillary electrophoresis (Agilent 5200). The particle size, zeta potential, and particle dispersion index (PDI) were determined using a Darwin analyzer. Encapsulation efficiency was calibrated using the Quant-iT™ RiboGreen RNA kit. The sizes of the mRNA-LNPs ranged from 90 to 110 nm.

Sample	Encapsulation Efficiency	Integrality	Particle Size (nm)	Zeta Potential	PDI (≤0.2)	dsRNA
CVG199 LNP	95	90.1	92.3	−2.4	0.17	≤0.40%
CVG200 LNP	96	91.5	90.8	−3.4	0.18	≤0.40%
CVG201 LNP	95	91.8	95.8	−2.8	0.20	≤0.40%
CVG202 LNP	94	91.8	95.3	−1.5	0.20	≤0.40%
CVG203 LNP	91	94.2	111.0	−6.4	0.25	≤0.40%
CVG204 LNP	93	91.1	97.9	−3.2	0.18	≤0.40%
CVG205 LNP	88	78.8	110.7	−3.6	0.22	≤0.40%
CVG206 LNP	98	90.6	91.9	−2.5	0.10	≤0.40%

## Data Availability

Data are contained within the article and Appendix A.

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
