# Peer review of "An mRNA Vaccine for Herpes Zoster and Its Efficacy Evaluation in Naïve/Primed Murine Models"

_vaccines, 2025, doi:10.3390/vaccines13030327_

Round 1
Reviewer 1 Report
Comments and Suggestions for Authors
In the manuscript entitled “A mRNA vaccine for Herpes Zoster and its efficacy evaluation in naïve/primed murine models,” the authors tested the hypothesis that truncated gEs may result in different immunogenicity. Accordingly, different mRNA sequences encoding antigens were designed. Humoral and cellular immune responses were evaluated systematically in naïve and VZV-primed mice models. It’s an exciting work due to the urgency to discover new treatments against Herpes zoster and effective vaccines to combat it with safe for the patient. Thus, for acceptance, I recommend a review of the English language and improving the introduction with information about some drugs used to treat the Herpes Zoster that can complement and justify the search for new vaccines, highlighting the advantages. In addition, the use of AlphaFold must be clear in the discussion and in the conclusion.
Comments on the Quality of English Languagerevise
Author Response
Comment 1: Regarding your concern about English editing
Response 1: We have reviewed the entire manuscript and improved the English writing.
Comment 2: Regarding your suggestion, “Introduction with information about some drugs used to treat Herpes Zoster that can complement and justify the search for new vaccines, highlighting the advantages.” We have done as follows:
Response 2: In the revised Introduction section, I have incorporated a new subsection discussing nucleic acid-based antiviral therapies (e.g., valacyclovir) and traditional prophylactic vaccines for VZV, including a detailed analysis of their respective advantages and limitations. Building upon these limitations, the text further highlights the technological superiority of mRNA vaccines over existing therapeutic and preventive approaches in terms of immunogenicity, safety, and clinical applicability. For specific revisions, please refer to the edited content in the Introduction section of the manuscript.
Comment 3: Regarding your suggestions,”Add more information on how we used Alphafold in the discussion and conclusion.” We have done as follows:
Response 3: Per your recommendations, I have incorporated a comprehensive discussion on the application and significance of AlphaFold into the Discussion and Conclusions sections of the manuscript. The added content highlights how AlphaFold’s structural predictions informed our analysis of immunogenicity variations across gE protein truncations and validated the functional relevance of key domains. Specific details, including the methodological integration and interpretive insights, are outlined below. (For more details, please check the discussion and conclusion sections in the manuscript.)
In the discussion section, we integrated “In this study, the integration of AlphaFold (version 2.3.2) prediction and the in vivo ELISpot evaluation in Naïve mice model provided critical insights into the structural basis of immunogenicity variations observed across truncated forms (CVG199 to CVG206) of the varicella-zoster virus (VZV) glycoprotein E (gE) (shown in Figure 4). By predicting the 3D structures of gE truncations (e.g., CVG199: 1–593 aa, CVG200: 1–573 aa, CVG201: 1–546 aa) and immunoblot-counting, AlphaFold explained the potential of a systematic correlation between structural domains and their functional contributions to cellular immunity. The study revealed that the C-terminal region (~546–623 aa), despite constituting only ~11% of the full-length gE protein, harbors structurally dynamic motifs that likely enhance the antigen presentation and T-cell epitope exposure. This observation aligns with the experimental data showing that the C-terminal truncations (e.g., CVG201: 1–546 aa) contributed disproportionately lower IL-2/IFN-γ responses (30.8% and 36.6%, respectively) compared to constructs retaining the C-terminal region (e.g., CVG199: 1–593 aa induced 88.8% IL-2 and 80% IFN-γ responses). Furthermore, AlphaFold’s structural predictions facilitated identifying potential conformational changes caused by truncations. For instance, removing residues 594–623 aa (as in CVG199) may disrupt stabilizing loops or sol-vent-exposed helices critical for MHC class II interactions, explaining the attenuated immunogenicity.”
In the conclusion section, we incorporated “By bridging structural predictions with functional immunogenicity assays; this study underscores AlphaFold’s and AI algorithm designs’ transformative role in accelerating anti-gen optimization for mRNA vaccines. The computational insights not only validated empirical findings but also provided a roadmap for future engineering of antigen-based vaccines to maximize cellular immunity against specific pathogen infections.”
Reviewer 2 Report
Comments and Suggestions for Authors
The authors report the development of a mRNA vaccine for Herpes Zoster, and as such the vaccine does induce humoral immune responses in a mouse model. Given that there is now a recombinant subunit vaccine that is highly effective and without any meaningful side-effects, it is unlikely that the mRNA vaccine would reach clinical utility.
Although the authors acknowledge that the mouse is not a model of zoster, it is not sufficient to intimate that their results are relevant to the human correlate. In that regard, results displaying results in the "primed" model are irrelevant, including analysis of splenic immune responses. The authors should consider revising their submission to report only the simplest results of their experiments, ie that they developed a mRNA construct that induced humoral responses, and deleting many of the sections and figures that are otherwise irrelevant. They would need to demonstrate in a Phase I/II trial in humans that their construct induces humoral immune responses, and that their construct is safe. There is no discussion of the potential side effects of mRNA vaccines, such as occurred with mRNA COVIC-19 vaccines, some being very serious.
Author Response
Dear Reviewer,
We sincerely appreciate your thorough evaluation of our manuscript and the insightful feedback you have provided. Below, we address each of your comments and recommendations point-by-point to ensure clarity in how we have incorporated your suggestions into the revised manuscript.
Comments 1: Regarding your statement, “Given that there is now a recombinant subunit vaccine that is highly effective and without any meaningful side-effect, it is unlikely that the mRNA vaccine would reach clinical utility.”The reply is as follows:
Responses 1: As mentioned in the Introduction section, while recombinant protein vaccines like Shingrix® have demonstrated commercial success and apparent clinical efficacy (with 97.2% effectiveness in preventing shingles among participants over 50 years old), they fail to fundamentally address patient challenges. The vaccine remains cost-prohibitive for developing nations, requiring approximately $500 for complete two-dose immunization in China. Furthermore, Shingrix® induces systemic adverse reactions (e.g., fever, myalgia) in over 30% of recipients and faces inherent limitations in production scalability, quality control, formulation stability, potential toxicity, and undesirable hemolytic effects associated with its QS21 adjuvant system.
Notably, certain immunocompromised patients experience symptom recurrence within six months post-vaccination despite completing the two-dose regimen, resulting in persistent mild postherpetic neuralgia (PHN). This clinical limitation stems from Shingrix's suboptimal cellular immunity, which fails to completely eliminate residual neurotoxic varicella-zoster virus (VZV) in hosts.
Our comparative analysis between CVG206 and Shingrix demonstrates distinct advantages of our mRNA vaccine platform. CVG206 exhibits significantly reduced adverse reactions while achieving superior cellular immunity - a hallmark strength of RNA vaccine technology. When combined with intradermal delivery optimization and lyophilization advancements, this platform shows potential for enhanced efficacy, substantially reduced production costs, and improved accessibility through price reduction.
These findings underscore the critical need to develop cost-effective, well-tolerated VZV mRNA vaccines with enhanced cellular immunogenicity, even in the current Shingrix era. The initiation of clinical trials for CVG206 represents a strategic advancement toward addressing these unmet medical needs.
Comments 2: Regarding your statement, “Although the authors acknowledge that the mouse is not a model of zoster, it is not sufficient to intimate that their results are relevant to the human correlate. In that regard, results displaying results in the “primed” model are irrelevant, including analysis of splenic immune responses.” and “they developed a mRNA construct that induced humoral responses, and deleting many of the sections and figures that are otherwise irrelevant.” The reply is as follows:
Responses 2: Establishing animal models for human-specific pathogens remains a significant scientific challenge. The rationale for employing a "primed model"—where varicella-zoster virus (VZV) is introduced 35 days prior to immunization—lies in its ability to recapitulate the natural history of human VZV infection. By mimicking the primary childhood infection, this model induces immune memory and alters both humoral and cellular immunity compared to naïve animal models, thereby better reflecting the clinical immune landscape in adults.
Notably, this primed model has been extensively validated and adopted by leading vaccine developers. For instance, GSK utilized it in the development of their subunit vaccine Shingrix® (Ref. 1), while Moderna and Greenlight Biosciences have similarly incorporated this approach (Ref. 2, 3). Given its widespread acceptance in the field, we adopted this model to evaluate our vaccine candidate, ensuring clinical relevance and alignment with industry standards.
Regarding the analysis of splenic immune responses and the omission of certain cellular immunity data, we emphasize that cellular immunity is particularly critical for eradicating latent viral reservoirs, such as those seen in VZV infection. Robust T-cell responses are essential for clearing neurotoxic virus remnants and preventing recurrence or complications like postherpetic neuralgia (PHN). While streamlining the manuscript necessitated removing some datasets, we retained key findings that directly demonstrate our vaccine’s superior cellular immunogenicity. Sharing these results provides readers with critical insights into the mechanistic advantages of our candidate and underscores its potential to address unmet clinical needs.
Reference
[1] Dendouga, N.; Fochesato, M.; Lockman, L.; Mossman, S.; Giannini, S.L. Cell-Mediated Immune Responses to a Varicella-Zoster Virus Glycoprotein E Vaccine Using Both a TLR Agonist and QS21 in Mice. Vaccine 2012, 30, 3126–3135, doi:10.1016/j.vaccine.2012.01.088.
[2] Monslow, M.A.; Elbashir, S.; Sullivan, N.L.; Thiriot, D.S.; Ahl, P.; Smith, J.; Miller, E.; Cook, J.; Cosmi, S.; Thoryk, E.; et al. Immunogenicity Generated by mRNA Vaccine Encoding VZV gE Antigen Is Comparable to Adjuvanted Subu-nit Vaccine and Better than Live Attenuated Vaccine in Nonhuman Primates. Vaccine 2020, 38, 5793–5802, doi:10.1016/j.vaccine.2020.06.062.
[3] Bhattacharya, A.; Jan, L.; Burlak, O.; Li, J.; Upadhyay, G.; Williams, K.; Dong, J.; Rohrer, H.; Pynn, M.; Simon, A.; et al. Potent and Long-Lasting Humoral and Cellular Immunity against Varicella Zoster Virus Induced by mRNA-LNP Vaccine. npj Vaccines 2024, 9, 72, doi:10.1038/s41541-024-00865-5.
Comments 3: Regarding your suggestion to “demonstrate in a Phase I/II trial in humans that their construct induces humoral immune responses, and their construct is safe.” My reply is as follows:
Responses 3:The Phase I/II trial holds pivotal significance in validating our vaccine candidate’s efficacy, as its therapeutic potential stems primarily from robust cellular immunity rather than humoral immune responses. While humoral immunity plays a critical role in primary prevention, it inadequately addresses the persistent challenge of postherpetic neuralgia (PHN) in patients.
Currently, we are advancing the Investigational New Drug (IND) application for CVG206 and intend to initiate Phase I/II clinical trials upon regulatory approval. To ensure successful execution, we are optimizing strategic resource allocation and budgeting for these upcoming studies. Furthermore, we remain actively seeking strategic partnerships to accelerate development and plan to disseminate our findings through peer-reviewed publications to contribute meaningfully to the field.
Comments 4: We fully understand your concerns regarding “the potential side effects of mRNA vaccines, such as occurred with mRNA COVID-19 vaccines, some being very serious.”. My reply is as follows:
Responses 4: To address this inquiry, we have incorporated an analysis of SARS-CoV-2 mRNA vaccine-associated adverse effects in the Introduction section (page 3). This addition systematically examines the etiology, mechanistic underpinnings, and technological solutions for these transient reactions. A representative summary we added in the introduction states:
“Although the novel SARS-CoV2 mRNA vaccines are associated with certain adverse effects, advancements in RNA vaccine technology hold significant promise for mitigating and potentially resolving these challenges. Importantly, the observed risks are acceptable trade-offs when weighed against the substantial advantages of this technology.”
Summary: Dear reviewer, overall, I believe vaccine researchers and developers should enhance their understanding of the critical role of cellular immunity. In pre-pandemic vaccine development, traditional paradigms overwhelmingly prioritized humoral immunity and antibody-dependent effects such as antibody-dependent cellular cytotoxicity (ADCC). This mindset has constrained innovation in vaccine design and target selection—a limitation rooted in both historical, technological constraints and the inherent challenges of inducing robust cellular immunity. Conventional vaccine platforms like inactivated vaccines and recombinant protein vaccines, while excelling at eliciting strong humoral responses, largely fail to induce meaningful cellular immunity. Consequently, most vaccine development has focused on preventing viral infections or reducing disease severity rather than effectively eliminating latent pathogens such as dormant viruses (e.g., VZV) or intracellular bacteria.
The post-pandemic era, however, has witnessed the rise of RNA vaccine technology, whose superior induction of cellular immunity now enables the development of therapeutic vaccines targeting latent infections. For pathogens like varicella-zoster virus (VZV), Mycobacterium tuberculosis, and Rickettsia species, RNA-based vaccines represent the optimal strategy to achieve both prophylactic and therapeutic clearance of persistent infections.
Reviewer 3 Report
Comments and Suggestions for Authors
Linglei Jiang et have developed an mRNA vaccine for Herpes Zoster and evaluated in naïve/primed murine models. Work is exciting and suitable for “Vaccine Journal”
This work can be accepted after addressing the following concerns;
- LNP/lyophilized LNP/Patch, please specify the formulation in the abstract for better understanding
- Please include the zeta potential of the formulation
- Include scale bar for TEM image
- It would be informative if the author could differentiate the bleb formation in LNPs referring to the TEM image
- Improve the quality of figure 1
- Line 379: correct statement; The polydispersity index (PDI) of mRNA-LNPs was smaller than 0.2, suggesting a heterogeneous distribution of these LNPs
- Table 1: Please mention the replicates and provide SD
- Figure 3E: the author also provides the log values for the post-transfection assay
Language is fine
Author Response
Dear Reviewer, We sincerely appreciate your thorough evaluation of our manuscript and the insightful feedback you have provided. Below, we address each of your comments and recommendations in a point-by-point manner to ensure clarity in how we have incorporated your suggestions into the revised manuscript.
Comments 1: Regarding your suggestion, “LNP/lyophilized LNP/Patch, please specify the formulation in the abstract for better understanding.”
Responses 1: At this stage, the specific formulation parameters of our lyophilized patch vaccine platform remain confidential due to ongoing regulatory evaluation processes with the CDE and FDA. While we regret our current inability to disclose these proprietary details, full methodological transparency will be provided in subsequent peer-reviewed publications following successful Investigational New Drug (IND) clearance and initiation of Phase I Varicella Zoster Virus (VZV) clinical trials.
Comments 2: “Please include the zeta potential of the formulation.”
Responses 2: The zeta potential information is in Table 2.
Comments 3: “Include scale bar for TEM image”.
Responses 3: A 50 nm scale bar was added to the left-bottom corner of the image.
Comments 4: “It would be informative if the author could differentiate the bleb formation in LNPs referring to the TEM image.”
Responses 4: While we acknowledge the seminal contributions of Pieter Cullis' groundbreaking theory on LNP architecture, significant debate persists regarding the correlation between bleb structures and optimal endosomal escape efficiency. Current literature presents compelling evidence that non-bleb LNP configurations may demonstrate superior transfection efficacy. To maintain scientific neutrality in our manuscript, we deliberately omitted theoretical comparisons in this controversial area. However, in response to your expert commentary, we have annotated potential bleb-like formations with white arrows in Figure 1E (TEM analysis) for your critical evaluation.
Comments 5: “Improve the quality of figure 1”.
Responses 5: We sincerely appreciate your valuable feedback. Per your recommendation, Figure 1 has been upgraded to a high-resolution format in the revised manuscript (see updated Figure 1).
Comments 6: “Line 379: correct statement; The polydispersity index (PDI) of mRNA-LNPs was smaller than 0.2, suggesting a heterogeneous distribution of these LNPs”.
Responses 6: The sentence has been corrected: "The mRNA-LNPs exhibited a polydispersity index (PDI) below 0.2, indicating a narrow size distribution characteristic of highly uniform nanoparticle populations.”
Comments 7: “Table 1: Please mention the replicates and provide SD”.
Responses 7: Thank you for asking about the replicates and SD in Table 1; however,for the particle size, zeta potential, and particle dispersion index (PDI) that were determined by a Darwin analyzer were only measured once for each mRNA-LNP. I agree that replicates and SD should be conducted on mRNA samples in our future study.
Comments 8: “Figure 3E: the author also provides the log values for the post-transfection assay.”
Responses 8: We have provided the log values for the post-transfection assay in Figure 3E. Please see the value table in the manuscript.
Round 2
Reviewer 2 Report
Comments and Suggestions for Authors
The authors have responded to some of the comments and suggestions, but cannot get beyond the fact that human trials are needed to establish safety and efficacy.
Author Response
Thank you for your feedback and comments. We wish to emphasize that our manuscript, explicitly titled "An mRNA Vaccine for Herpes Zoster and Its Efficacy Evaluation in Naïve/Primed Murine Models," represents a preclinical investigation focused on vaccine development using mRNA technology and immunological evaluation in murine models. The methodology and conclusions presented sufficiently validate our vaccine's immunogenic efficacy at the animal study level. Human clinical trials neither align with the stated research objectives nor are they required to substantiate the scientific merits of this discovery-phase study.
Reviewer 3 Report
Comments and Suggestions for Authors
Authors have implemented all the corrections.
Author Response
Thank you for your professional comments and suggestions